# Retrievals of heavy ozone with MIPAS

Bastiaan Jonkheid[1], Thomas Röckmann[1], Norbert Glatthor[2], Christof Janssen[3], Gabriele Stiller[2], and Thomas von Clarmann[2]

[1]Institute for Marine and Atmospheric Research, Utrecht University, Utrecht, the Netherlands
[2]Karlsruher Institut für Technologie, Institut für Meteorologie und Klimaforschung, Karlsruhe, Germany
[3]LERMA-IPSL, Sorbonne Universités, UPMC Univ. Paris 6, Observatoire de Paris, PSL Research University, CNRS, Paris, France

**Abstract.** A method for retrieval of $^{18}$O-substituted isotopomers of $O_3$ in the stratosphere with the Michelson Interferometer for Passive Atmospheric Sounding (MIPAS) is presented. Using a smoothing regularisation constraint, volume mixing ratio profiles are retrieved for the main isotopologue and the symmetric and asymmetric isotopomers of singly substituted $O_3$. For the retrieval of the heavy isotopologues, two microwindows in the MIPAS A band (685-970 cm$^{-1}$) and seven in the AB band (1020-1170 cm$^{-1}$) are used. As the retrievals are performed as perturbations on the previously retrieved a priori profiles, the vertical resolution of the individual isotopomer profiles is very similar, which is important when taking the ratio between two isotopomers. The performance of the method is evaluated using 1044 vertical profiles recorded with MIPAS on 1 July 2003. The mean values are separated by latitude band, along with estimates for their systematic uncertainties. The asymmetric isotopomer shows a mean enrichment of $\sim 8\%$, with a vertical profile that increases up to 33 km and decreases at higher altitudes. This decrease with altitude is a robust result that does not depend on retrieval settings, and it has not been reported clearly in previously published datasets. The symmetric isotopomer is considerably less enriched, with mean values around 3% and with a large spread. In individual retrievals the uncertainty of the enrichment is dominated by the measurement noise (2–4%), which can be reduced by averaging multiple retrievals; systematic uncertainties linked to the retrieval are generally small at 1–2%. The variabilities in the retrieval results are largest for the southern hemisphere.

## 1 Introduction

The abundance of the heavy oxygen isotopes $^{17}$O and $^{18}$O in stratospheric $O_3$ is unusually large compared to the ambient $O_2$. This isotopic enrichment was first discovered for $^{18}$O by in-situ mass spectrometer measurements by Mauersberger (1981). Laboratory experiments by Thiemens and Heidenreich (1983) found that the isotopic enrichment does not follow the common mass-dependent fractionation law. Rather, enrichments of $^{17}$O and $^{18}$O are of almost equal magnitude. This unique isotopic signature, called the ozone isotope effect, is a useful tracer for the role of $O_3$ in atmospheric chemistry, and research into its exact causes is still ongoing. Since the discovery over thirty years ago, these findings have been confirmed by further mass spectrometer measurements of atmospheric samples (Krankowsky et al., 2007), and by high-resolution spectroscopy using surface-based total-column measurements (Rinsland et al., 1985; Meier and Notholt, 1996), balloon-borne instruments (Abbas et al., 1987; Goldman et al., 1989; Johnson et al., 2000; Haverd et al., 2005), and space-based spectrometers (Irion et al.,

1996; Piccolo et al., 2009; Sato et al., 2014). Laboratory research indicated that the observed isotopic enrichment is primarily controlled by unusually strong isotope effects associated with the $O_3$ formation reaction (Morton et al., 1990):

$$O + O_2 \leftrightharpoons O_3^* \xrightarrow{M} O_3 \qquad (R1)$$

Here, M is a bath gas molecule that stabilises the excited complex $O_3^*$. It was also found that the enrichment is mainly located in the asymmetric isotopomers (Mauersberger et al., 1993). The rate coefficients of many individual isotopic combinations in reaction (R1) have been measured (Anderson et al., 1997; Janssen et al., 1999; Mauersberger et al., 1999), and it was found that there is a strong dependence of the rate coefficient of (R1) on the change in zero-point energy in the equivalent exchange reaction $O + O_2 \rightarrow O_3^* \rightarrow O_2 + O$ (Janssen et al., 2001). Theoretical calculations suggest that the enrichment of the asymmetric isotopomers can likely be explained by a non-statistical behaviour of the excited complex, making it more stable and less likely to spontaneously dissociate (Hathorn and Marcus, 1999; Gao and Marcus, 2002).

In addition to the ozone formation reaction, the photolysis of ozone in the reaction

$$O_3 + h\nu \rightarrow O_2 + O \qquad (R2)$$

also causes an enrichment of the heavy isotopologues by preferential destruction of the lighter ones. Initial evidence for this comes from laboratory experiments (Bhattacharya and Thiemens, 1988; Chakraborty and Bhattacharya, 2003), and was recently substantiated by measurements where photolytic $O_3$ removal could be analytically separated from chemical removal (Früchtl et al., 2015a). Wavelength-dependent isotope effects in $O_3$ photolysis are also suggested by theoretical calculations (Miller et al., 2005; Liang et al., 2006; Ndengué et al., 2014), but the agreement of these calculations with new high-precision measurements is relatively poor (Früchtl et al., 2015b). Nevertheless, isotope effects in photolysis have been invoked as explanation for the high enrichments found in the upper stratosphere by balloon measurements (Haverd et al., 2005; Krankowsky et al., 2007).

The isotopic composition is commonly reported in delta notation, where the heavy-to-light isotope ratio $R$ in a sample is compared to the same ratio in a reference material. The enrichment $\delta$ of $^kO$ ($k = 17, 18$) is then defined as

$$\delta\left(^kO\right) = \left(\frac{R\left(^kO\right)_{obs}}{R\left(^kO\right)_{ref}} - 1\right) \qquad (1)$$

Since the isotope effects for $O_3$ are very large, they are often reported in percent (%) rather than per mill, and this convention is followed in the present paper. The primary international reference ratio $R\left(^nO\right)_{ref}$ is Vienna Standard Mean Ocean Water (VSMOW; $R\left(^{17}O\right)_{VSMOW} = 7.799 \times 10^{-4}$, $R\left(^{18}O\right)_{VSMOW} = 2.00520 \times 10^{-3}$), but in studies of the $O_3$ isotopic composition the ratio of atmospheric $O_2$ is often used. Atmospheric $O_2$ is enriched by 1.208% in $^{17}O$ and by 2.388% in $^{18}O$ with respect to VSMOW (Barkan and Luz, 2005), and this is the standard used throughout this paper.

The enrichments derived in this paper are calculated for singly $^{18}O$-substituted $O_3$, i.e. an ozone molecule consisting of two $^{16}O$ isotopes and one $^{18}O$ isotope. The isotopologues are labeled by their total mass, so that $^{48}O_3$ denotes an ozone molecule consisting of three $^{16}O$ isotopes and $^{50}O_3$ an ozone molecule with two $^{16}O$ isotope and one $^{18}O$ isotope. The enrichment of $^{50}O_3$ is then calculated from the ratio $R_{obs}\left(^{50}O_3\right) = \left[^{50}O_3\right] / \left(3\left[^{48}O_3\right]\right)$. The statistical factor 3 in the denominator accounts

for the three positions where the $^{18}$O atom can reside in the molecule, and allows the observed ratio to be compared to the reference, which is defined as the atomic ratio. Where applicable, the symmetric and asymmetric isotopomers of $^{50}$O$_3$ are indicated with the prefix s or a, i.e. s$-^{50}$O$_3$ denotes $^{16}$O$^{18}$O$^{16}$O and a$-^{50}$O$_3$ denotes $^{18}$O$^{16}$O$^{16}$O. Enrichments of a$-^{50}$O$_3$ include a statistical factor of 2 in the denominator of $R_{\mathrm{obs}}$.

5    In this paper, satellite retrievals of $\delta(\mathrm{a}-^{50}\mathrm{O}_3)$ and $\delta(\mathrm{s}-^{50}\mathrm{O}_3)$ derived from the MIPAS instrument on Envisat are presented. The purpose is to investigate whether and how the MIPAS observations can be used to provide independent data on ozone enrichment in the middle and upper stratosphere, and to detect and quantify possible temporal and geographical variations. For validation, these enrichments are compared to previously published results from the balloon-based thermal emission FTIR spectra of Johnson et al. (2000), the balloon-based solar FTIR spectra of Haverd et al. (2005), the space-based ATMOS solar spectra of Irion et al. (1996) and the space-based SMILES thermal emission spectra by Sato et al. (2014); the total enrichment of $^{50}$O$_3$ is compared to the presumably highest precision mass spectrometer data of Krankowsky et al. (2007). The previously published infrared spectroscopy datasets show comparatively large variations between measurements, and neither Irion et al. (1996) nor Johnson et al. (2000) report significant variation with latitude or altitude, or any seasonal trends. Haverd et al. (2005) find a pronounced vertical trend that is attributed to photolytic enrichment. For two balloon flights, anomalously low enrichments are associated with a high stratospheric aerosol content due to the eruption of Mount Pinatubo. Sato et al. (2014) report a correlation between $\delta(\mathrm{a}-^{50}\mathrm{O}_3)$ and temperature that is consistent with experimental data. The mass spectrometer data of Krankowsky et al. (2007) show far less variation between measurements; the reported enrichments in the upper stratosphere are ascribed to photolytic enrichment, while a seasonal trend at high latitudes can be explained by temperature differences.

This paper is structured as follows: the description of the data and the retrieval procedure, including the error analysis, is given in Section 2. The results are discussed in Section 3, and conclusions are presented in Section 4.

## 2    Data and retrieval

### 2.1    MIPAS

The Michelson Interferometer for Passive Atmospheric Sounding (MIPAS) is a limb-sounding spectrometer on the Envisat satellite. It sampled the mid-infrared region between 685 and 2410 cm$^{-1}$ with a spectral resolution of 0.025 cm$^{-1}$ during the first two years of observations (from June 2002 to March 2004). Due to problems with the interferometer mirror slide, the instrument operated on a reduced spectral resolution of 0.0625 cm$^{-1}$ after this period; operations ended with the loss of contact with Envisat in April 2012. An overview of MIPAS and its capabilities is given by Fischer et al. (2008).

There exist several processors that retrieve ozone volume mixing ratios from MIPAS level 1b data; an overview and inter-comparison is given by Laeng et al. (2015). These algorithms differ in the microwindows and regularisation technique used, but yield similar results. In this work, the retrieval processor developed by KIT-IMK (Karlsruhe) and IAA/CSIC (Granada) is used. A detailed description is given in von Clarmann et al. (2003); its performance on ozone retrievals was studied in detail by Glatthor et al. (2006). Validation studies performed by Steck et al. (2007) and Laeng et al. (2014) showed that MIPAS ozone

retrievals have realistic error estimates, and suffer from only small biases of 5%–20% (depending on altitude and reference instrument).

MIPAS data have been used to retrieve the ozone isotopologues $^{49}O_3$ and $^{50}O_3$ by Piccolo et al. (2009). There was generally a good agreement between the MIPAS-Envisat-derived enrichments and those obtained from FIRS-2 (Johnson et al., 2000)
and MIPAS-B (Friedl-Vallon et al., 2004) for a$-^{50}O_3$ and s$-^{50}O_3$; the enrichments for $^{49}O_3$ were higher than expected from either theory or FIRS-2 observations. In the current paper, the latter isotopologue is not examined because its weak spectral lines makes it hard to quantify accurately; rather, special attention is given to the enrichment profiles of $^{50}O_3$ for different latitude bands.

## 2.2  Retrieval set-up

To retrieve the volume mixing ratios (VMRs) of the different ozone isotopologues contained in the atmospheric state vector $x$ from the measured irradiances $y$, the retrieval processor iteratively minimises the least squares equation (von Clarmann et al., 2003):

$$\chi^2 = \left(y - f(x)\right)^T S_y^{-1} \left(y - f(x)\right) + \left(x - x_a\right)^T R \left(x - x_a\right) \tag{2}$$

where $f$ is the forward radiative transfer model, $S_y$ is the error covariance matrix of the observations, $x_a$ is the a priori state
vector and $R$ is the regularisation matrix. This procedure is structurally similar to to the scheme described by Rodgers (2000). The forward model is the Karlsruhe Optimised and Precise Radiative transfer Algorithm (KOPRA; Stiller et al., 2002). The regularisation matrix is of the form $R = T_1^T \Lambda T_1 + D$, where $T_1$ is a first-order derivative operator and $\Lambda$ is a diagonal matrix containing weights for each altitude step (cf. Tikhonov, 1963; Steck, 2002). This term maps the shape (but not the values) of the a priori $O_3$ profile to the heavy $O_3$ profile when the measurement does not resolve the full structure of the latter. $D$ is a
diagonal matrix which is usually zero everywhere except for the elements referring to the topmost altitudes of 100 and 120 km; it forces the ozone VMRs at these altitudes to be close to zero. The a priori values $x_a$ are MIPAS IMK/IAA V5H ozone profiles, assuming $\delta(^{50}O_3) = 0$. Terms that account for continuum radiation and radiance offset due to calibration errors are retrieved along with the atmospheric state vector $x$.

Instead of investigating the full MIPAS spectrum for each retrieval, a microwindow approach is used to analyse only the
spectral regions that supply information on $^{48}O_3$ and $^{50}O_3$ abundances (cf. von Clarmann and Echle, 1998). A set of nine microwindows is used, based on the set of Piccolo et al. (2009) with slight modifications; this set contains two microwindows in the A band (685–970 cm$^{-1}$) and seven in the AB band (1020–1170 cm$^{-1}$), see Table 1. Although including the AB band may lead to biases in retrievals of $^{48}O_3$ (see Laeng et al., 2014, 2015, for more information), the AB band contains many suitable emission lines for the retrieval of $^{50}O_3$. By excluding the AB band both a$-^{50}O_3$ and s$-^{50}O_3$ are limited to three
degrees of freedom (defined as the trace of the averaging kernel, see Rodgers, 2000), while including the AB band increases this number to six. The precision of the retrievals also suffers when the AB band data are disregarded, with standard deviations that are a factor of four larger. Thus, the gain in isotopic information by including the spectral lines in the MIPAS AB band outweighs the problem of the possible bias between the A and AB bands.

From the resulting VMRs of $^{48}O_3$, $a-^{50}O_3$ and $s-^{50}O_3$ the isotopic enrichments $\delta(a-^{50}O_3)$ and $\delta(s-^{50}O_3)$ are calculated. The total enrichment, used for comparison with mass spectrometer data, is then given by $\delta(^{50}O_3) = (2\delta(a-^{50}O_3) + \delta(s-^{50}O_3))/3$. The relevant isotopomers are retrieved using an existing MIPAS ozone product as a priori profiles, appropriately scaled in the case of $^{50}O_3$; thus, the results can be interpreted as a perturbation on these a priori profiles. This means that matching the vertical resolution of the different isotopomers, normally required to avoid mixing of information from different altitudes, is not an important issue. The regularisation parameters were chosen in such a way that

- The precision of the resulting $\delta(^{50}O_3)$ profiles is optimised. In practice this means that the regularisation is strong enough that it results in smooth vmr profiles.

- The number of degrees of freedom of the $^{50}O_3$ retrievals is maximised, given the above restriction.

- The vertical resolutions of the $^{48}O_3$ retrievals are similar to those of the heavy isotopomers; vertical resolution is defined here as the full width at half maximum of the averaging kernels. This means that the $^{48}O_3$ profiles are regularised more strongly than would be necessary for an independent retrieval, but this way the ratio of the VMRs makes physical sense while the sensitivity loss reported by Worden et al. (2012) is avoided.

## 2.3 Error analysis

With the retrieval set-up in place, the uncertainties in the profiles of the VMRs and their covariances are calculated. The following sources of uncertainties in the retrievals are considered:

- Measurement noise.

- Uncertainties in the profiles of interfering species. Gases considered here are $H_2O$, $CO_2$, $N_2O$, $CH_4$, $SO_2$, $NO_2$, $NH_3$, $HNO_3$, $HCN$, $C_2H_2$, $C_2H_6$, $COF_2$, CFC-11, CFC-12, CFC-22, $SF_6$, CFC-113, $N_2O_5$, $ClONO_2$ and $CH_3COCH_3$.

- Uncertainties in the temperature profile, and in horizontal temperature gradients.

- Uncertainties in the spectroscopic data in the HITRAN database.

- Uncertainties in the instrumental parameters: line of sight, residual spectral shift, gain calibration, and instrumental line shape.

The uncertainties due to these parameters are calculated for 128 example geolocations, distributed over five latitude bands: polar (N), 60°N– 90°N; mid-latitude (N), 30°N–60°N; tropical, 30°S–30°N; mid-latitude (S), 30°S–60°S; and polar (S), 60°S–90°S. The calculations were performed on retrievals for July 1, 2003; 16 for daytime retrievals and 16 for nighttime retrievals except in the polar latitude bands.

The errors in the VMRs due to measurement noise are given by

$$\mathbf{S}_{\text{noise}} = \mathbf{G}\mathbf{S}_y\mathbf{G}^T \tag{3}$$

where $\mathbf{G}$ is the gain matrix $\mathbf{G} = (\mathbf{K}_x^T \mathbf{S}_y^{-1} \mathbf{K}_x + \mathbf{R})^{-1} \mathbf{K}_x^T \mathbf{S}_y^{-1}$, and $\mathbf{K}_x$ is the Jacobian $\mathbf{K}_x = d\boldsymbol{y}/d\boldsymbol{x}$.

The errors due to retrieval parameter uncertainties are given by

$$\mathbf{S}_{\text{parameter}} = \mathbf{G} \mathbf{K}_b \mathbf{S}_b \mathbf{K}_b^T \mathbf{G}^T \qquad (4)$$

where $\mathbf{K}_b$ is the Jacobian $\mathbf{K}_b = d\boldsymbol{y}/d\boldsymbol{b}$ and $\boldsymbol{b}$ is a vector containing the retrieval parameters, with associated uncertainties $\mathbf{S}_b$.

The total VMR error is

$$\mathbf{S}_{\text{VMR}} = \mathbf{S}_{\text{noise}} + \mathbf{S}_{\text{parameter}} \qquad (5)$$

The errors in the derived enrichments of $s -^{50}O_3$ are given by (cf. Steinwagner et al., 2007)

$$\mathbf{S}_{\delta(s -^{50}O_3)} = \mathbf{J}_s \mathbf{S}_s \mathbf{J}_s^T \qquad (6)$$

where for a given altitude level $i$:

$$\mathbf{J}_{s,i} = \left( \frac{\partial \delta(s -^{50}O_3)_i}{\partial \text{VMR}(^{48}O_3)_i}, \frac{\partial \delta(s -^{50}O_3)_i}{\partial \text{VMR}(s -^{50}O_3)_i} \right) \qquad (7)$$

$$\mathbf{S}_s = \begin{pmatrix} \mathbf{S}_{^{48}O_3} & \mathbf{S}_{^{48}O_3, s -^{50}O_3} \\ \mathbf{S}_{s -^{50}O_3, ^{48}O_3} & \mathbf{S}_{s -^{50}O_3} \end{pmatrix} \qquad (8)$$

Here, the matricies $\mathbf{S}_{^{48}O_3}$ and $\mathbf{S}_{s -^{50}O_3}$ are the subsets of $\mathbf{S}_{\text{VMR}}$ that contain the covariance matrices of $\text{VMR}(^{48}O_3)$ and $\text{VMR}(s -^{50}O_3)$, and the matrix $\mathbf{S}_{^{48}O_3, s -^{50}O_3} = \mathbf{S}_{s -^{50}O_3, ^{48}O_3}^T$ accounts for correlations between the two VMRs. A similar expression is used for $a -^{50}O_3$.

# 3   Results and discussion

## 3.1   Uncertainty analysis

The measurement uncertainties are shown in Figure 1. It can be seen that for each individual retrieval the measurement noise is the dominant source of uncertainty, yielding an absolute uncertainty in the enrichment of about 3% in most latitude bands at altitudes where ozone is present in significant amounts. Measurement noise is lowest in the northern hemisphere and tropics; in the southern latitude bands, particularly the polar band, it is higher at 4–5%. The uncertainty due to retrieval parameters is smaller than the measurement noise, at 1–2%; this component of the uncertainty is dominated by the uncertainty in the vertical CFC-113 profile, and by uncertainties in instrument gain calibration and line of sight. It should be noted that uncertainties in the HITRAN spectroscopic data for $^{50}O_3$ may be underestimated, particularly due to the difficulty to distinguish between the symmetric and asymmetric isotopomers (Janssen, 2005). The uncertainties all increase sharply in all latitude bands at altitudes above 50 km. While the uncertainty due to measurement noise can be reduced by averaging over a large number of profiles, the uncertainties in retrieval parameters, particularly instrumental parameters, will typically remain.

## 3.2 Retrieved enrichments

For all of the 1044 geolocations covered during July 1 2003, the vmr profiles of $^{48}O_3$, a$-^{50}O_3$ and s$-^{50}O_3$ are retrieved. Of these, 66 are disregarded because the retrieval did not converge. Of the remaining 978 profiles, the altitude levels where the diagonal of the averaging kernel is less than a threshold value of 0.03 are disregarded because there is virtually no information

for that level. On average, the retrievals have six degrees of freedom, with a vertical resolution of 6 km between 25 and 50 km of altitude.

The profiles of the mean total enrichments derived from these data are shown in Figure 2, separated by latitude band. The enrichments show values between 5 and 10% at altitudes between 20 and 50 km. The shape of the enrichment profiles is consistent over the northern and tropical latitude bands, and still discernible in the southern mid-latitude band: there is a steady

increase from 25 to 35 km, a decrease until 40 km, and an increase again until 48 km. The standard deviation of the retrievals at 25–50 km is generally small (4-5%) in the tropics and in the northern hemisphere, but much larger (up to 10%) in the southern hemisphere; the fact that the standard deviations are much larger than the estimated accuracy suggests that this is due to zonal variability in these latitude bands. In all latitude bands, the standard deviations tend to increase at altitudes <25 km and >50 km, as do retrieval parameter uncertainties in Figure 1.

Global means for the enrichments of a$-^{50}O_3$ and s$-^{50}O_3$, and for total $^{50}O_3$ are shown in Figure 3. The enrichment of the asymmetric isotopomer shows the same vertical trends as the total enrichment in Figure 2. The standard deviation of the zonal means is relatively small and a factor of 2–4 larger than the uncertainty of the retrieved parameters. The symmetric isotopomer shows a very different vertical structure: at altitudes below 30 km the enrichment is around 5–8%, of the same order as the asymmetric isotopomer, with a maximum at 27 km. This maximum is primarily caused by high enrichments at

northern latitudes. A similar maximum is visible in the standard deviation, but this originates from the larger variability in the southern hemisphere (cf. Figure 4). At higher altitudes the enrichment is almost constant at a lower value of a few percent, and has lower standard deviations. The vertical trends of the total global enrichment profile resemble those of the northern latitude profiles in Figure 2, and the enrichment profile of the asymmetric isotopomer, although the maximum at 35 km is less pronounced due to averaging with the southern hemisphere profiles that do not show this feature.

A more detailed look at the enrichment of both the asymmetric and symmetric isotopomers of $^{50}O_3$ as a function of latitude and altitude is given in Figure 4. The enrichments of a$-^{50}O_3$ vary between 5 and 15%, with a clear vertical structure that is visible at all latitudes north of $40°S$. This vertical structure is similar to that of $\delta(^{50}O_3)$ shown in Figure 2, with an increasing enrichment to 33 km, followed by a decrease to 40 km, and then a further increase until 47 km. The standard deviation in the enrichments is a few percent at most altitudes. At latitudes south of $40°S$, the vertical structure is less pronounced, and standard

deviations increase to 8–10%. Under these conditions, little can be said about the vertical structure of the enrichment.

It can be seen that s$-^{50}O_3$ has enrichments of only a few percent at all latitudes and altitudes. Isotopic depletions occur at some places. Since there is no known mechanism that could cause these depletions in ozone, it is likely a retrieval artefact, possibly due to a spectroscopic bias. There is a maximum in the enrichment around 27 km in the northern hemisphere. Standard deviations are generally 5% or higher, and reach their maximum of 20–40% between $50°$ and $80°S$.

The vertical structure of the total enrichment combines the characteristics of the two isotopomers; it resembles that of $a-^{50}O_3$, although the lower maximum is smeared out due to the lower altitude peak of $\delta(s-^{50}O_3)$, while the second maximum is at 50 km rather than 47 km due to the sharp increase in the $\delta(s-^{50}O_3)$ profile there.

## 3.3 Comparison with previously published data

### 3.3.1 By latitude band

In the northern hemisphere there is generally good agreement between the MIPAS-derived enrichments and previously published data (see Figure 2). In the polar latitude band there is a slight discrepancy between MIPAS and the mass spectrometer data of Krankowsky et al. (2007), but this is within the uncertainty limits of both datasets. Interestingly, the MIPAS data show a decrease at high altitudes ($> 35$ km) that is not seen in other datasets.

In order to exclude that this decrease is due to an artefact of the retrieval procedure, several tests were performed:

- The vertical extent of the microwindows was expanded in order to increase the altitude range in which information is gathered. In particular, the A-band microwindows were extended up to 47 km in altitude. This did not result in a better agreement of the retrievals with previous independent data, and increased the number of retrievals with convergence problems.

- The altitude up to which a continuum term is fitted to the observations was increased from the previous value of 33 km. The physical reason for this is the possible presence of aerosols in the middle and upper stratosphere (Neely et al., 2011). Some of the oscillations seen in the enrichment profiles decreased in amplitude by setting the cut-off for continuum regularisation to 50 km; this value is used in the retrievals presented in this paper. Further increasing this altitude did not improve the retrievals and again caused convergence problems.

- A two-step retrieval was used with a different regularisation strategy. In the first step, only $^{48}O_3$ is retrieved. In the second step, only $a-^{50}O_3$ was retrieved, using as a priori $a-^{50}O_3$ profile the result of the $^{48}O_3$ retrieval, scaled by $R_{ref}$, to which an enrichment of 10% was added. The regularisation matrix was limited to the matrix $\mathbf{D}$, with full diagonal values, while the vertical derivative term was set to zero. In this way, the regularisation procedure forces enrichment profiles to be close to a constant 10%. The resulting enrichments showed a similar vertical structure as before, albeit with more noise since the resulting profiles are no longer smoothed in the regularisation.

The fact that the decrease above 35 km is retrieved in all these tests is reason to conclude that it is inherent in the MIPAS observations rather than an issue with the retrievals.

In the southern hemisphere the retrieved enrichments show comparatively large standard deviations. At mid-latitudes there is a good agreement with the ATMOS-derived enrichments up to an altitude of 35 km. Higher in the atmosphere the MIPAS enrichments decrease, while the ATMOS enrichments increase with height. At southern polar latitudes the MIPAS enrichments are lower than those of ATMOS at all altitudes. The standard deviations in the results are very large for both datasets, and there is a large overlap within one standard deviation.

### 3.3.2 Global

At altitudes below 33 km the global mean total enrichments of MIPAS are in good agreement with previous observations (cf. Figure 3), particularly with the mass spectrometer data of Krankowsky et al. (2007). The disagreement with other datasets that is visible in the tropical and northern hemisphere latitude bands persists in the global average.

The global mean enrichments of $a-^{50}O_3$ derived by MIPAS are generally in good agreement with previous observations. As is the case for total $^{50}O_3$ in the tropical and northern latitude bands, discussed above, the MIPAS global mean $\delta(a-^{50}O_3)$ shows a decrease above 33 km; other datasets also show a decrease with altitude, starting at 40 km for the FTIR dataset of Haverd et al. (2005) and the SMILES dataset of Sato et al. (2014), or at 30 km for the FIRS-2 dataset of Johnson et al. (2000). The ATMOS dataset of Irion et al. (1996) shows a steady increase up until the maximum retrieval altitude of 40 km. Furthermore,

a decreasing enrichment above 35 km is also predicted by photochemical models, caused by the large fractionation in the long-wavelength tail of the Hartley band (Liang et al., 2006; Ndengué et al., 2014). These models predict maxima at 35 km with amplitudes of 3% for $a-^{50}O_3$ and 6% for $s-^{50}O_3$.

The symmetric isotopomer profiles show large standard deviations below 30 km and above 45 km, making comparison with other datasets problematic at those altitudes. Between 30 and 45 km, the MIPAS enrichments are lower than those of the FTIR

and ATMOS datasets; where the latter show the enrichments increasing with altitude, the former is steady or shows a slight decrease.

### 4   Conclusions

This paper presents MIPAS observations of the enrichments of $^{18}O$ in stratospheric ozone for July 1 2003. The retrieval set-up discussed here has projected accuracies in the enrichments of $< 2\%$ at all latitudes, and at most altitudes between 20 and 50 km;

it should be noted that the effect of uncertainties in the spectroscopic data is likely underestimated, so this figure is probably an underestimate. For individual retrievals, measurement noise is the main source of uncertainty, between 2 and 4% under most conditions but much higher (up to 8%) at southern polar latitudes.

The mean MIPAS enrichments agree well with previous balloon-based and space-based observations at low altitudes ($< 35$ km). At higher altitudes the MIPAS enrichments show a clear decrease. While other datasets also have decreasing enrichments

with increasing altitude beyond some point, this decrease is usually much weaker and occurs at a different altitude. This feature is unlikely to be a retrieval artefact, and is probably inherent in the MIPAS data; it may be caused by photolytic effects, as suggested by theoretical models. The large standard deviations of the retrievals in the southern hemisphere, with a magnitude several times larger than the estimated accuracy, indicate variability in the profiles within a latitude band. The enrichment of the asymmetric isotopomer is typically higher ($\delta \sim 8 - 10\%$) than that of the symmetric isotopomer ($\delta \sim 2 - 3\%$).

In conclusion, isotope information can be obtained with acceptable precision from MIPAS measurements. Although systematic errors, in particular from spectroscopic uncertainties, are hard to quantify, such systematic effects are expected to remain constant in time, and likely also in space. Therefore, MIPAS isotope data can be used to study the spatial and temporal variability of the isotopic composition of stratospheric $O_3$.

*Acknowledgements.* The authors want to thank Vanessa Haverd, Bill Irion, David Johnson and Tomohiro Sato for providing their measurements of heavy ozone enrichment. This work was funded by the Netherlands Organisation for Scientific Research as project ALW-GO/12/05, Isotope measurements from space on stratospheric ozone (ISOZONE). The authors acknowledge provision of MIPAS level-1b data by ESA.

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

**Table 1.** Microwindows used in this study

| Spectral range ($cm^{-1}$) | Vertical extent (km) |
|---|---|
| 713.800 – 714.500 | 18 – 30 |
| 960.950 – 963.950 | 18 – 36 |
| 1020.725 – 1023.650 | 6 – 68 |
| 1037.825 – 1038.375 | 52 – 68 |
| 1042.125 – 1042.875 | 47 – 68 |
| 1043.225 – 1044.675 | 6 – 39 |
| 1053.975 – 1055.350 | 18 – 68 |
| 1088.000 – 1091.000 | 6 – 68 |

**Figure 1.** The uncertainties in the retrievals of $\delta^{50}O_3$. Shown are the total noise uncertainty and the total uncertainty due to parameters in the retrieval set-up, together with its three principal sources: uncertainties in the CFC-113 profile, in the line of sight, and in gain calibration.

**Figure 2.** Total enrichments of $^{50}O_3$, separated by latitude band. The mean enrichments are shown by solid lines, the total uncertainty due to retrieval parameters are given by dotted lines, and the standard deviations are given by the shaded regions. Shown are the data retrieved in the current paper, and the mass spectrometer data of Krankowsky et al. (2007), the FTIR data of Haverd et al. (2005), the FIRS-2 data of Johnson et al. (2000) and the ATMOS data of Irion et al. (1996). The previously published data are shown on a 5 km-resolution grid, where the means and standard deviations are calculated for each altitude bin.

**Figure 3.** Globally averaged enrichments of a$-^{50}$O$_3$, s$-^{50}$O$_3$ and total $^{50}$O$_3$. As in Figure 2, the mean enrichments are shown by solid lines, the total uncertainty due to retrieval parameters by the dotted lines, and the standard deviations by the shaded regions. Presented are the data retrieved in the current paper and previously measured values (see the caption of Figure 2 for references), the latter of which are shown on a 5 km grid. Also shown are the SMILES data of Sato et al. (2014), which is on its own grid of approximately 5 km vertical resolution.

**Figure 4.** The mean retrieved enrichments of $^{50}O_3$ as a function of latitude and altitude (left column), and the standard deviation of the enrichments (right column). The upper and middle rows represent the asymmetric and symmetric isotopomers, respectively, and the lower row shows the total enrichment. The contours indicate 10% intervals.