# Peer review of "Retrievals of heavy ozone with MIPAS"

_Atmospheric Measurement Techniques, 2016_

## Referee Comment (RC1) · Anonymous Referee #1 · 8 Jul 2016

The paper by Jonkheid et al. presents the retrieval of Ozone isotopomers from MIPAS instrument. This is an interesting subject presented in a well written and concise manuscript. The results seem to be robust and should constitute a valuable contribution to AMT after consideration of the following comments.

In general, the paper would benefit from a more detailed description of the error and sensitivity of the retrieval. Also it is not clear to me how such measurements could be used to investigate the role of O3 in atmospheric chemistry. It is my understanding that it is not the purpose of the paper to expand on this subject, but a few lines, references, would strengthen the interest of developing such products.

[Figure]

**Specific comments**

Interest of such products

The authors state that observations of $O_3$ isotopomers are "useful tracer for the role of $O_3$ in atmospheric chemistry" (P1,L20). Is it possible to be more explicit on their use, is there any former work using such observations?

Later on, it is suggested that ozone formation and photolysis enrich delta values. I guess then, delta observations could help to constrain these processes. Is there any uncertainty associated to these processes?

Uncertainty analysis 3.1

The error analysis in is actual form is a very general description of the error analysis and more information would be needed to have a clearer picture on the robustness of the results.

- What are the parameter uncertainties used in the error calculation (a table would be appropriate)? For example, we can not appreciate any potential underestimation of the uncertainties associated to HITRAN (P6, L23) since theses uncertainties are not given.

- A clearer decomposition in random and systematic error would also be useful. Especially since the different impacts of these errors is discussed later on.

- Is there a reason why there is no description of the smoothing error? At least a discussion on the sensitivity of the retrieval is needed. For the moment, the discussion is limited to the information that there are 6 DFS between 25 and 50 km.

- Figures of the averaging kernels would greatly help to actually realize that the different species have very similar vertical resolution (also for all latitudinal bands). Is there any sensitivity below 25 km? That would be useful information for the following of the manuscript.

- P7,L12: " the fact that the standard deviations are much larger than the estimated accuracy suggests that this is due to zonal variability in these latitude bands". It could also be due to an underestimation of the error. Or differences in sensitivity? Note that it is the random error here it is about and not the accuracy.

- In the Polar (S) panel, the error estimation appears to be the lowest, but in the error estimation corresponding to this latitudinal band, the error is the largest. Is it because more profiles have been used? Could you add the information on the number of profiles. Also if you attribute the standard deviations to natural variability, you need to explain the depletion observed for example in Mid-latitude.

- "In all latitude bands, the standard deviations tend to increase at altitudes <25 km and >50 km, as do retrieval parameter uncertainties in Figure 1." What is the meaning of this sentence? What is the confidence level in the retrieved values below 25 km and above 45 km? It is needed to precise whether or not at theses altitudes, the retrieval reflects real variations of O3 isotopomers.

- The lack of discussion on error sources also reflect in the explanation concerning the depletion observed in s-50O3 (P7,L32) as you suggest that the "depletion to be a retrieval artefact possibly due to a spectroscopic bias". I would expect the error associated with a spectroscopic bias to be systematic and constant over time and space (as it is stipulated in the conclusions), so why this is just happening here? Moreover in the abstract the systematic uncertainty is estimated to 1-2 % but the observed depletion actually seems larger than this estimation.

Comparison with previously published data 3.3

This section in its actual form is a bit difficult to follow for someone not familiar with all the existing O3 measurements. A description of the other dataset is missing. A short description of the others published data, their sampling, sensitivity and error characteristics would facilitate the reading. Why do the different instruments present such different standard deviations? Is it inherent to their errors? I guess the different datasets cover different time period, could that be an eventual cause of the spread observed in some cases?

---

## Referee Comment (RC2) · Anonymous Referee #2 · 21 Jul 2016

Retrievals of Heavy Ozone with MIPAS by Jonkheid et al

Reviewer Comments.

Summary

The paper describes profile retrievals of the relative concentrations of 18O variants of ozone, based on a single day of MIPAS data. The authors acknowledge and appear to deal adequately with the two major problems associated with 'ratio' retrievals: the handling of systematic error correlations and differences in vertical resolution/sampling. It is unfortunate that the analysis is limited to just a single day as I would have liked to have seen if the observed distributions were reproducible from day-to-day, or if there is any hint of a seasonal cycle. Perhaps that will be a separate publication.

But, as it stands, the paper is clear and concise, it presents scientifically interesting and justified results, and I have only minor comments and queries, as listed below.

Minor comments

P2, L24: Suggest 'parts per thousand' rather than 'parts per mill' in case the non-expert reader assumes that 'mill' is an abbrevation for 'million'.

P2, L27: I believe HITRAN implicitly assumes VSMOW isotopic concentrations by scaling the line intensities. I suppose the authors have taken this into account. This could also slightly modify the a priori assumption of deltaˆ50 O_3 = 0 (P4, L22) unless the standard IMK/IAA retrieval also modifies the isotopic intensities.

P4, Retrieval Set-Up. I eventually concluded that the retrieval here is a combined retrieval of the main isotopologue as well as both minor isotopomers, rather than just a retrieval of the minor isotopomers assuming the IMK/IAA V5 ozone for the major isotopologue. It could be made a little clearer that this is the case (or, if it isn't, then I don't understand P5,L10).

P4, L27: Table 1 lists only 6 microwindows in the AB band.

P4, L27: Given that most of the O3 absorption is in the AB band, I suppose the question here is whether it is really necessary to include the two A band microwindows? Given the altitude range for these two microwindows, their inclusion appears to be for low altitude ozone where, presumably, the AB lines are saturated.

P6 Eq (7): It took a while before I understood how this fitted in with Steinwagner's formulation, but then I realised that this was only the generic definition of J. Perhaps it would help to add a further equation showing the evaluation of these two terms given the definition of delta.

P6, L13: 'matricies' should be 'matrices'. Strictly speaking these cross-terms don't account for the 'correlation between the two VMRs' (which, I assume, a priori are assumed to be independent) but the error correlations in the retrievals. On this I assume that all the systematic errors are assumed fully correlated across all microwindows/altitudes?

P6, L18: I was initially surprised how small the error was in the 'enrichment' retrieval given that 5% would be considered a 'good' uncertainty for a retrieval of the main isotopologue. But then reading more carefully I realised that this actually represented an 'absolute' uncertainty in the enrichment, which is itself only a few percent. Perhaps add something here and to the Figure 1 caption to point this out, and/or add curves showing 'typical' retrieved enrichment profiles to the panels for comparison.

P6, L23: It seems odd that, of all the interfering molecules, CFC-113 should provide the most significant uncertainty. It is a minor atmospheric component (20 pptv in the troposphere, much less in the stratosphere), a long-lived gas (so one would expect its profile to be relatively well modelled) and a heavy molecule with broad spectral features (so the continuum retrieval should remove much of the effect). Is it just an unfortunate overlap for a particular key O3 isotopic line?

P6, L23: How was the gain uncertainty modelled? If it is assumed fully correlated (spectrally) one would expect the error to almost disappear once the ratio of minor/major isotopologues is taken since it just acts as a scaling of the spectrum.

P6, L27: Even instrumental parameter errors (associated with calibration) might be expected to reduce with averaging, but the spectroscopic uncertainty is the classic case of an error which (a) poorly defined and (b) irreducible by averaging.

P7, L1: 'vmr' - upper case used elsewhere

P7, L15: 'Global mean' can be calculated in a variety of ways from a simple average of all available data at whatever latitude, to cosine(latitude) weighted averages allowing for the different area of each latitude band. Given the sparsity of some of the comparison data I guess it the former is used. However, making such a comparison does assume that either (a) the latitudinal variation is small compared with the difference in datasets (which may be true), or (b) all datasets sample the range of latitudes with equal weight (which is certainly not the case)

L9, P19: '<2% accuracy' - I regard accuracy as the sum of the random error (precision) plus systematic errors, but this figure <2% seems to represent just the systematic error, unless the authors are referring to the accuracy of an averaged product rather than individual profiles. Either way, clarification is required.

---

## Author Comment (AC1) · 22 Aug 2016

**1   Interest of such products**

- $O_3$ affects the isotopic composition of other atmospheric species, and this is one of the reasons why the ozone isotope effect is studied. It is beyond the scope of the current paper to go into much detail here, but several affected species ($CO_2$, CO and $N_2O$) are referenced in the revised version.

- The uncertainties of the processes behind the isotopic enrichment of $O_3$ are difficult to quantify in terms of enrichments occurring in the atmosphere. The fractionation of the formation reaction rate coefficient, with its dependence on pressure and temperature, is quite well known, but this does not translate directly to an enrichment value. Conversely, while there are some data on the photolytic effect, there is only poor agreement between laboratory measurements and theoretical

calculations. The observational evidence for photolytic enrichment is tentative. In the revised manuscript, values are given for the magnitude of both effects, and their uncertainties are described in qualitative terms.

**2   Uncertainty analysis 3.1**

- In the revised manuscript, the uncertainties in the retrieval parameters are shown in the new Table 2. While there are too many relevant $^{50}O_3$ lines in the HITRAN database to list them all in the paper, note that all have the uncertainty flag "0" (meaning a relative uncertainty >1 or unreported) for line position, intensity and air-pressure induced line shift. This is also stated explicitly.

- The discussion of the uncertainties is changed in the revised version to give a clearer distinction between random and systematic error sources. Most notably, Figure 3 (Figure 2 in the original manuscript) is changed so that the total uncertainty is shown, instead of the component due to retrieval parameters.

- The most significant vertical variation of all relevant isotopemers is assumed to be captured in the a priori profiles, and the profiles retrieved here show only slowly varying variations on the prior data. For this reason, the smoothing error is not considered to be an important source of uncertainty; this is stated in the revised version.

- A new figure was added to the manuscript showing the averaging kernels of the relevant isotopomers (Figure 2).

- The discussion of the uncertainties is changed in the revised version. In the northern and tropical latitude bands, the enstimated precision fits quite well with the observed standard deviation; in the southern latitude bands, it is noted that the observed variation is larger than the estimated precision.

- The number of retrievals used to obtain the means and standard deviations in each latitude band is now displayed in the figure. The negative enrichments needed to explain the large spread in the observed profiles are noted explicitly.

- The passage is rephrased in the revised manuscript. The spread in the observed profiles is now compared directly to the estimated precision, which makes it easier to see if the variation is naturally occurring or inherent in the retrieval process.

- A spectroscopic bias resulting in a constant shift would indeed make sense. A positive shift of a few percent would remove the negative enrichments found here, this is noted in the revised manuscript.

**3  Comparison this previously published data 3.3**

- The description of the other datasets has been moved from the introduction to its own Section 3.3.1. The description of the measurement techniques, the error characteristics (where available) and the resulting profiles is expanded in the revised manuscript.

  Please also note the supplement to this comment:
  http://www.atmos-meas-tech-discuss.net/amt-2016-144/amt-2016-144-AC1-supplement.pdf

**Supplement:**

**Retrievals of heavy ozone with MIPAS**

[revised manuscript text omitted]

Ocean Water (VSMOW; $R\left(^{17}\mathrm{O}\right)_{\mathrm{VSMOW}} = 7.799 \times 10^{-4}$, $R\left(^{18}\mathrm{O}\right)_{\mathrm{VSMOW}} = 2.00520 \times 10^{-3}$), but in studies of the $\mathrm{O}_3$ isotopic composition the ratio of atmospheric $\mathrm{O}_2$ is often used. Atmospheric $\mathrm{O}_2$ is enriched by 1.208% in $^{17}\mathrm{O}$ and by 2.388% in $^{18}\mathrm{O}$ with respect to VSMOW (Barkan and Luz, 2005), and this is the standard used throughout this paper.

The enrichments derived in this paper are calculated for singly $^{18}\mathrm{O}$-substituted $\mathrm{O}_3$, i.e. an ozone molecule consisting of two $^{16}\mathrm{O}$ isotopes and one $^{18}\mathrm{O}$ isotope. The isotopologues are labeled by their total mass, so that $^{48}\mathrm{O}_3$ denotes an ozone molecule consisting of three $^{16}\mathrm{O}$ isotopes and $^{50}\mathrm{O}_3$ an ozone molecule with two $^{16}\mathrm{O}$ isotope and one $^{18}\mathrm{O}$ isotope. The enrichment of $^{50}\mathrm{O}_3$ is then calculated from the ratio $R_{\mathrm{obs}}\left(^{50}\mathrm{O}_3\right) = \left[^{50}\mathrm{O}_3\right]/\left(3\left[^{48}\mathrm{O}_3\right]\right)$. The statistical factor 3 in the denominator accounts for the three positions where the $^{18}\mathrm{O}$ atom can reside in the molecule, and allows the observed ratio to be compared to the reference, which is defined as the atomic ratio. Where applicable, the symmetric and asymmetric isotopomers of $^{50}\mathrm{O}_3$ are indicated with the prefix s or a, i.e. $\mathrm{s}-^{50}\mathrm{O}_3$ denotes $^{16}\mathrm{O}^{18}\mathrm{O}^{16}\mathrm{O}$ and $\mathrm{a}-^{50}\mathrm{O}_3$ denotes $^{18}\mathrm{O}^{16}\mathrm{O}^{16}\mathrm{O}$. Enrichments of $\mathrm{a}-^{50}\mathrm{O}_3$ include a statistical factor of 2 in the denominator of $R_{\mathrm{obs}}$.

In this paper, satellite retrievals of $\delta(\mathrm{a}-^{50}\mathrm{O}_3)$ and $\delta(\mathrm{s}-^{50}\mathrm{O}_3)$ derived from the MIPAS instrument on Envisat are presented. The purpose is to investigate whether and how the MIPAS observations can be used to provide independent data on ozone enrichment in the middle and upper stratosphere, and to detect and quantify possible temporal and geographical variations. For validation, these enrichments are compared to previously published results from the balloon-based thermal emission FTIR spectra of Johnson et al. (2000), the balloon-based solar FTIR spectra of Haverd et al. (2005), the space-based ATMOS solar spectra of Irion et al. (1996) and the space-based SMILES thermal emission spectra by Sato et al. (2014); the total enrichment of $^{50}\mathrm{O}_3$ is compared to the presumably highest precision mass spectrometer data of Krankowsky et al. (2007).

This paper is structured as follows: the description of the data and the retrieval procedure, including the error analysis, is given in Section 2. The results are discussed in Section 3, and conclusions are presented in Section 4.

**2  Data and retrieval**

**2.1  MIPAS**

The Michelson Interferometer for Passive Atmospheric Sounding (MIPAS) is a limb-sounding spectrometer on the Envisat satellite. It sampled the mid-infrared region between 685 and 2410 cm$^{-1}$ with a spectral resolution of 0.025 cm$^{-1}$ during the first two years of observations (from June 2002 to March 2004). Due to problems with the interferometer mirror slide, the instrument operated on a reduced spectral resolution of 0.0625 cm$^{-1}$ after this period; operations ended with the loss of contact with Envisat in April 2012. An overview of MIPAS and its capabilities is given by Fischer et al. (2008).

There exist several processors that retrieve ozone volume mixing ratios from MIPAS level 1b data; an overview and inter-comparison is given by Laeng et al. (2015). These algorithms differ in the microwindows and regularisation technique used, but yield similar results. In this work, the retrieval processor developed by KIT-IMK (Karlsruhe) and IAA/CSIC (Granada) is used. A detailed description is given in von Clarmann et al. (2003); its performance on ozone retrievals was studied in detail by Glatthor et al. (2006). Validation studies performed by Steck et al. (2007) and Laeng et al. (2014) showed that MIPAS ozone

retrievals have realistic error estimates, and suffer from only small biases of 5%–20% (depending on altitude and reference instrument).

MIPAS data have been used to retrieve the ozone isotopologues $^{49}O_3$ and $^{50}O_3$ by Piccolo et al. (2009). There was generally a good agreement between the MIPAS-Envisat-derived enrichments and those obtained from FIRS-2 (Johnson et al., 2000) and MIPAS-B (Friedl-Vallon et al., 2004) for $a-^{50}O_3$ and $s-^{50}O_3$; the enrichments for $^{49}O_3$ were higher than expected from 5 either theory or FIRS-2 observations. In the current paper, the latter isotopologue is not examined because its weak spectral lines makes it hard to quantify accurately; rather, special attention is given to the enrichment profiles of $^{50}O_3$ for different latitude bands.

**2.2 Retrieval set-up**

To retrieve the volume mixing ratios (VMRs) of $^{48}O_3$, $a-^{50}O_3$ and $s-^{50}O_3$, contained in the atmospheric state vector $\boldsymbol{x}$, from 10 the measured irradiances $\boldsymbol{y}$, the retrieval processor iteratively minimises the least squares equation (von Clarmann et al., 2003):

$$\chi^2 = (\boldsymbol{y} - \boldsymbol{f}(\boldsymbol{x}))^T \mathbf{S}_y^{-1} (\boldsymbol{y} - \boldsymbol{f}(\boldsymbol{x})) + (\boldsymbol{x} - \boldsymbol{x}_a)^T \mathbf{R} (\boldsymbol{x} - \boldsymbol{x}_a) \tag{2}$$

where $\boldsymbol{f}$ is the forward radiative transfer model, $\mathbf{S}_y$ is the error covariance matrix of the observations, $\boldsymbol{x}_a$ is the a priori state vector and $\mathbf{R}$ is the regularisation matrix. This procedure is structurally similar to to the scheme described by Rodgers (2000). 15 The forward model is the Karlsruhe Optimised and Precise Radiative transfer Algorithm (KOPRA; Stiller et al., 2002). The regularisation matrix is of the form $\mathbf{R} = \mathbf{T}_1^T \boldsymbol{\Lambda} \mathbf{T}_1 + \mathbf{D}$, where $\mathbf{T}_1$ is a first-order derivative operator and $\boldsymbol{\Lambda}$ is a diagonal matrix containing weights for each altitude step (cf. Tikhonov, 1963; Steck, 2002). This term maps the shape (but not the values) of the a priori $O_3$ profile to the heavy $O_3$ profile when the measurement does not resolve the full structure of the latter. $\mathbf{D}$ is a diagonal matrix which is usually zero everywhere except for the elements referring to the topmost altitudes of 100 and 120 20 km; it forces the ozone VMRs at these altitudes to be close to zero. The a priori values $\boldsymbol{x}_a$ come from MIPAS IMK/IAA V5H ozone profiles. For $^{48}O_3$ these exact profiles are used, while for $^{50}O_3$ they are scaled according to $\delta(^{50}O_3) = 0$ with respect to the standard used in the HITRAN database; in terms of the standard adopted in the present paper, this is a prior enrichment of $-3.02\%$. Terms that account for continuum radiation and radiance offset due to calibration errors are retrieved along with the atmospheric state vector $\boldsymbol{x}$.

25 Instead of investigating the full MIPAS spectrum for each retrieval, a microwindow approach is used to analyse only the spectral regions that supply information on $^{48}O_3$ and $^{50}O_3$ abundances (cf. von Clarmann and Echle, 1998). A set of eight microwindows is used, based on the set of Piccolo et al. (2009) with slight modifications; this set contains two microwindows in the A band (685–970 cm$^{-1}$) and six in the AB band (1020–1170 cm$^{-1}$), see Table 1. Although including the AB band may lead to biases in retrievals of $^{48}O_3$ (see Laeng et al., 2014, 2015, for more information), the AB band contains many suitable 30 emission lines for the retrieval of $^{50}O_3$. By excluding the AB band both $a-^{50}O_3$ and $s-^{50}O_3$ are limited to three degrees of freedom (defined as the trace of the averaging kernel, see Rodgers, 2000), while including the AB band increases this number to six. The precision of the retrievals also suffers when the AB band data are disregarded, with standard deviations that are a

factor of four larger. Thus, the gain in isotopic information by including the spectral lines in the MIPAS AB band outweighs the problem of the possible bias between the A and AB bands. Conversely, the A band includes many $s-^{50}O_3$ lines, so this band cannot be ignored without a significant loss of information.

From the resulting VMRs of $^{48}O_3$, $a-^{50}O_3$ and $s-^{50}O_3$ the isotopic enrichments $\delta(a-^{50}O_3)$ and $\delta(s-^{50}O_3)$ are calculated. The total enrichment, used for comparison with mass spectrometer data, is then given by $\delta(^{50}O_3) = (2\delta(a-^{50}O_3) + \delta(s-^{50}O_3))/3$. The relevant isotopomers are retrieved using an existing MIPAS ozone product as a priori profiles, appropriately scaled in the case of $^{50}O_3$; thus, the results can be interpreted as a perturbation on these a priori profiles. This means that matching the vertical resolution of the different isotopomers, normally required to avoid mixing of information from different altitudes, is not an important issue. The regularisation parameters were chosen in such a way that

– The precision of the resulting $\delta(^{50}O_3)$ profiles is optimised. In practice this means that the regularisation is strong enough that it results in smooth vmr profiles.

– The number of degrees of freedom of the $^{50}O_3$ retrievals is maximised, given the above restriction.

– The vertical resolutions of the $^{48}O_3$ retrievals are similar to those of the heavy isotopomers; vertical resolution is defined here as the full width at half maximum of the averaging kernels. This means that the $^{48}O_3$ profiles are regularised more strongly than would be necessary for an independent retrieval, but this way the ratio of the VMRs makes physical sense while the sensitivity loss reported by Worden et al. (2012) is avoided.

**2.3 Error analysis**

With the retrieval set-up in place, the uncertainties in the profiles of the VMRs and their covariances are calculated. The following sources of uncertainties in the retrievals are considered:

– Measurement noise, based on instrumental parameters.

– Uncertainties in the profiles of interfering species. Gases considered here are $H_2O$, $CO_2$, $N_2O$, $CH_4$, $SO_2$, $NO_2$, $NH_3$, $HNO_3$, HCN, $C_2H_2$, $C_2H_6$, $COF_2$, CFC-11, CFC-12, CFC-22, $SF_6$, CFC-113, $N_2O_5$, $ClONO_2$ and $CH_3COCH_3$. The uncertainties are taken from MIPAS retrievals of these species.

– Uncertainties in the temperature profile, and in horizontal temperature gradients. The temperature is a retrieved quantity, and the uncertainty in the retrieval is used here. For the horizontal gradient, a fixed value is assumed (see Table 2).

– Uncertainties in the spectroscopic data in the HITRAN database. It should be noted that for $a-^{50}O_3$ and $s-^{50}O_3$ several critical parameters, such as line position and intensity, have no uncertainties given in the HITRAN database at the frequencies considered here. This component of the systematic uncertainty is therefore underestimated in the calculations.

– Uncertainties in the instrumental parameters: line of sight, residual spectral shift, gain calibration, and instrumental line shape. The assumed values for these uncertainties are given in Table 2.

A further source of uncertainty is the smoothing error, which accounts for the difficulty to retrieve small-scale variations. Because most of the vertical variation in the VMR profiles is already contained in the a priori information, and the focus is on the enrichment profiles which are expected to show only limited variations, the smoothing error is not considered significant here. Similarly, errors due to non-local thermal equilibrium effects are assumed to be irrelevant for the altitudes considered in this paper.

The uncertainties due to these parameters are calculated for 128 example geolocations, distributed over five latitude bands: polar (N), $60°$N–$90°$N; mid-latitude (N), $30°$N–$60°$N; tropical, $30°$S–$30°$N; mid-latitude (S), $30°$S–$60°$S; and polar (S), $60°$S–$90°$S. The calculations were performed on retrievals for July 1, 2003; 16 for daytime retrievals and 16 for nighttime retrievals except in the polar latitude bands. Following Glatthor et al. (2006), all errors are assumed to decrease when averaging a large number of profiles, except those due to uncertainties in line spectroscopic data and instrumental line shape.

The errors in the VMRs due to measurement noise are given by

$$\mathbf{S}_{\text{noise}} = \mathbf{G}\mathbf{S}_y\mathbf{G}^T \tag{3}$$

where $\mathbf{G}$ is the gain matrix $\mathbf{G} = (\mathbf{K}_x^T\mathbf{S}_y^{-1}\mathbf{K}_x + \mathbf{R})^{-1}\mathbf{K}_x^T\mathbf{S}_y^{-1}$, and $\mathbf{K}_x$ is the Jacobian $\mathbf{K}_x = d\mathbf{y}/d\mathbf{x}$.

The errors due to retrieval parameter uncertainties are given by

$$\mathbf{S}_{\text{parameter}} = \mathbf{G}\mathbf{K}_b\mathbf{S}_b\mathbf{K}_b^T\mathbf{G}^T \tag{4}$$

where $\mathbf{K}_b$ is the Jacobian $\mathbf{K}_b = d\mathbf{y}/d\mathbf{b}$ and $\mathbf{b}$ is a vector containing the retrieval parameters, with associated uncertainties $\mathbf{S}_b$.

A further source of uncertainty is the smoothing error, which accounts for the difficulty to retrieve small-scale variations. Because most of the vertical variation in the VMR profiles is already contained in the a priori information, and the focus is on the enrichment profiles which are expected to show only limited variations, the smoothing error is not considered significant here.

The total VMR error is then

$$\mathbf{S}_{\text{VMR}} = \mathbf{S}_{\text{noise}} + \mathbf{S}_{\text{parameter}} \tag{5}$$

The errors in the derived enrichments of $s - {}^{50}O_3$ are given by (cf. Steinwagner et al., 2007)

$$\mathbf{S}_{\delta(s-{}^{50}O_3)} = \mathbf{J}_s\mathbf{S}_s\mathbf{J}_s^T \tag{6}$$

where for a given altitude level $i$:

$$\mathbf{J}_{s,i} = \left( \frac{\partial\delta(s-{}^{50}O_3)_i}{\partial\text{VMR}({}^{48}O_3)_i}, \frac{\partial\delta(s-{}^{50}O_3)_i}{\partial\text{VMR}(s-{}^{50}O_3)_i} \right) = \left( \frac{-\text{VMR}(s-{}^{50}O_3)_i}{R({}^{18}O)_{\text{ref}}\,\text{VMR}({}^{48}O_3)_i^2}, \frac{1}{R({}^{18}O)_{\text{ref}}\,\text{VMR}({}^{48}O_3)_i} \right) \tag{7}$$

$$\mathbf{S}_s = \begin{pmatrix} \mathbf{S}_{{}^{48}O_3} & \mathbf{S}_{{}^{48}O_3,s-{}^{50}O_3} \\ \mathbf{S}_{s-{}^{50}O_3,{}^{48}O_3} & \mathbf{S}_{s-{}^{50}O_3} \end{pmatrix} \tag{8}$$

Here, the matrices $\mathbf{S}_{^{48}O_3}$ and $\mathbf{S}_{s-^{50}O_3}$ are the subsets of $\mathbf{S}_{\text{VMR}}$ that contain the covariance matrices of $\text{VMR}(^{48}O_3)$ and $\text{VMR}(s-^{50}O_3)$, and the matrix $\mathbf{S}_{^{48}O_3,s-^{50}O_3} = \mathbf{S}_{s-^{50}O_3,^{48}O_3}^T$ accounts for error correlations in the retrievals. A similar expression is used for $a-^{50}O_3$.

**3   Results and discussion**

**3.1   Uncertainty analysis**

The absolute measurement uncertainties, expressed in the units of the enrichments (i.e. as percentages), are shown in Figure 1. It can be seen that for each individual retrieval the measurement noise is the dominant source of uncertainty, yielding an absolute uncertainty in the enrichment of 2–4% in most latitude bands at altitudes where ozone is present in significant amounts. Measurement noise is lowest in the northern hemisphere and tropics; in the southern latitude bands, particularly the polar band, it is higher at 4%. The random uncertainty due to retrieval parameters is smaller than the measurement noise, at 1–2%; this component of the uncertainty is dominated by the uncertainty in the vertical CFC-113 profile, and by uncertainties in the line of sight. The uncertainty due to overlap of CFC-113 spectral features is likely overestimated; these features are nearly constant in each microwindow, and any uncertainty would be compensated for by the fitting of continuum radiation. The total random uncertainty, and thus overall precision, is around 4%, dependent on latitude and altitude.

The systematic errors caused by uncertainties in the spectroscopic parameters and the instrumental line shape are generally small at 0.5%. However, it should be noted that, in the microwindows considered here, the magnitude of the uncertainties in the spectroscopic data for $^{50}O_3$ are not given in the HITRAN database for several key parameters, such as line position and intensity. This is likely caused by the difficulty to distinguish between the symmetric and asymmetric isotopomers, noted by Janssen (2005). Therefore, this uncertainty is underestimated here.

[revised manuscript text omitted]

It can be seen that $s-^{50}O_3$ has enrichments of only a few percent at all latitudes and altitudes. Isotopic depletions occur at some places; as noted before, there is no known physical process that can cause these depletions, but a constant offset due to spectroscopic bias is possible. There is a maximum in the enrichment around 27 km in the northern hemisphere. Standard deviations are generally 5% or higher, and reach their maximum of 20–40% between $50°$ and $80°S$.

The vertical structure of the total enrichment combines the characteristics of the two isotopomers; it resembles that of $a-^{50}O_3$, although the lower maximum is smeared out due to the lower altitude peak of $\delta(s-^{50}O_3)$, while the second maximum is at 50 km rather than 47 km due to the sharp increase in the $\delta(s-^{50}O_3)$ profile there.

**3.3 Comparison with previously published data**

**3.3.1 Datasets**

The isotopic composition of stratospheric $O_3$ has been studied extensively since its discovery. The most comprehensive datasets, used here for validation of the MIPAS data, are the space-based solar absorption spectra by the ATMOS Fourier-transform infrared spectrometer (Irion et al., 1996); the balloon-borne thermal emission spectra by the FIRS-2 Fourier-transform spectrometer (Johnson et al., 2000); the balloon-borne solar absorption spectra by the MkIV FTIR spectrometer (Haverd et al., 2005); the mass spectrometer data of Krankowsky et al. (2007); and the space-borne emission spectra by the SMILES submillimetre sounder (Sato et al., 2014).

The ATMOS dataset consists of observations from the Spacelab-3 mission in July 1973 and the Atlas-1, -2 and -3 missions in March 1992, April 1993 and November 1994. It shows global average enrichments of $12 \pm 6\%$ for $a-^{50}O_3$ and $7\% \pm 7$ for $s-^{50}O_3$ (using the isotopic standard adopted in this paper), with no discernible variations with latitude or altitude. The uncertainties indicate the standard deviation in the measurements.

The FIRS-2 dataset uses data of seven ballon flights between 1989 and 1997, launched from Fort Sumner, Daggett and Fort Wainwright. The mean enrichments between 25 and 35 km of altitude are $12.2 \pm 1.0\%$ for $a-^{50}O_3$ and $6.1 \pm 1.8\%$ for $s-^{50}O_3$. Note that the uncertainties given here are the standard error of the mean as opposed to the standard deviation, which is typically larger. No significant variation with altitude is found.

The FTIR dataset uses data of seven balloon flights between 1997 and 2003, launched from Fort Wainwright, Esrange and Fort Sumner. Mean enrichments of $13.8 \pm 2.9\%$ and $7.0 \pm 2.6\%$ are reported for $a-^{50}O_3$ and $s-^{50}O_3$, respectively; it was not specified in the original publication whether these uncertainties denote standard deviations or standard errors. The absolute precision of the enrichments due to uncertainties in the retrieval parameters (temperature, pressure broadening coefficients, zero offset and slant column integration) is estimated to be around 2%. A pronounced vertical trend is attributed to photolytic enrichment. For two balloon flights, anomalously low enrichments are associated with a high stratospheric aerosol content due to the eruption of Mount Pinatubo.

The mass spectrometer data were collected from 11 balloon flights from Kiruna, Aire-sur-l'Adour and Teresina between 1998 and 2005. These measurements show the highest precision of the datasets considered here; at high altitudes an absolute precision of 0.4% in the enrichment is reported. The use of mass spectrometry makes it impossible to distinguish $a-^{50}O_3$ and $s-^{50}O_3$, however. The measurements at mid-latitude and equatorial latitudes show vertical trends that cannot be explained by the temperature effect in the formation reaction alone, and are ascribed to photolytic enrichment. The polar data show a seasonal trend that is consistent with the temperature.

The SMILES dataset was gathered at the International Space Station in February and March 2010. Only latitudes between $20°N$ and $40°N$ were considered. It contains enrichments only for the $a-^{50}O_3$ isotopomer, but it covers higher altitudes (up to 52 km) than the other datasets. The absolute systematic error in the enrichment varies between 4% and 6% at altitudes between 25 and 45 km. This includes contributions of uncertainties in the air-broadening parameter and its temperature dependence, in

the line intensity, in the antenna beam pattern, in the sideband separator characteristics and in the antenna response function. The vertical variation of the enrichment is found to be linked with variations in temperature.

**3.3.2 By latitude band**

[revised manuscript text omitted]

**Figure 3.** Total enrichments of $^{50}O_3$, separated by latitude band. The mean enrichments are shown (solid lines), as well as the total uncertainty in the retrievals (both random and systematic, dotted lines), and the standard deviations (shaded regions). The number of retrievals used in each band is indicated in the upper left corner. Also shown are the mass spectrometer data of Krankowsky et al. (2007), the FTIR data of Haverd et al. (2005), the FIRS-2 data of Johnson et al. (2000) and the ATMOS data of Irion et al. (1996). The previously published data are shown on a 5 km-resolution grid, where the means and standard deviations are calculated for each altitude bin.

[Figure]

**Figure 4.** Averaged enrichments of a$-^{50}O_3$, s$-^{50}O_3$ and total $^{50}O_3$. As in Figure 3, the mean enrichments are shown by solid lines, the total uncertainty by the dotted lines, and the standard deviations by the shaded regions. Presented are the data retrieved in the current paper and previously measured values (see the caption of Figure 3 for references), which are shown on a 5 km grid. Also shown are the SMILES data of Sato et al. (2014), which is on its own grid of approximately 5 km vertical resolution.

[Figure]

**Figure 5.** The mean retrieved enrichments of $^{50}O_3$ as a function of latitude and altitude (left column), and the standard deviation of the enrichments (right column). The upper and middle rows represent the asymmetric and symmetric isotopomers, respectively, and the lower row shows the total enrichment. The contours indicate 10% intervals.